# Microsoft Azure Kinect Calibration for Three-Dimensional Dense Point Clouds and Reliable Skeletons

**DOI:** 10.3390/s22134986

**Published:** 2022-07-01

**Authors:** Laura Romeo, Roberto Marani, Anna Gina Perri, Tiziana D’Orazio

**Affiliations:** 1National Research Council of Italy (CNR), Institute of Intelligent Industrial Technologies and Systems for Advanced Manufacturing (STIIMA), Via Amendola 122 D/O, 70126 Bari, Italy; laura.romeo@stiima.cnr.it (L.R.); tiziana.dorazio@stiima.cnr.it (T.D.); 2Department of Electrical and Information Engineering (DEI), Polytechnic University of Bari, Via Orabona 4, 70126 Bari, Italy; annagina.perri@poliba.it

**Keywords:** Azure Kinect, calibration, computer vision, point cloud, skeleton

## Abstract

Nowadays, the need for reliable and low-cost multi-camera systems is increasing for many potential applications, such as localization and mapping, human activity recognition, hand and gesture analysis, and object detection and localization. However, a precise camera calibration approach is mandatory for enabling further applications that require high precision. This paper analyzes the available two-camera calibration approaches to propose a guideline for calibrating multiple Azure Kinect RGB-D sensors to achieve the best alignment of point clouds in both color and infrared resolutions, and skeletal joints returned by the Microsoft Azure Body Tracking library. Different calibration methodologies using 2D and 3D approaches, all exploiting the functionalities within the Azure Kinect devices, are presented. Experiments demonstrate that the best results are returned by applying 3D calibration procedures, which give an average distance between all couples of corresponding points of point clouds in color or an infrared resolution of 21.426 mm and 9.872 mm for a static experiment and of 20.868 mm and 7.429 mm while framing a dynamic scene. At the same time, the best results in body joint alignment are achieved by three-dimensional procedures on images captured by the infrared sensors, resulting in an average error of 35.410 mm.

## 1. Introduction

In recent years, the need for trustworthy RGB-D sensors has increased importance in many fields, such as environment reconstruction for robotic applications [1,2,3], people tracking from healthcare systems [4,5], object recognition for manufacturing tasks [6,7,8], and gesture recognition for natural human–computer interfaces [9,10].

Among RGB-D devices, the Microsoft Azure Kinect [11] (Redmond, Washington, US), released in 2019, is a Time-of-Flight (ToF) sensor [12] that offers considerably higher accuracy than other commercially available devices [13] at low cost. It makes the Azure Kinect one of the most reliable cameras used in many research fields [14,15,16]. As an example, in [14], a hand gesture recognition system is proposed, where the depth image obtained with an Azure Kinect is used to obtain information about the joints of the hand. This information helps to develop a real-time system that defines and recognizes gestures indicating left, right, up, and down on a tabletop holographic display. In [16], an Azure Kinect is used to assess the possibility of anterior cruciate ligament rupture and osteoarthritis. In addition, the possibility of exploiting the Azure Kinect Software Development Kit (SDK), even for the extraction of skeletal joints with the Azure Kinect Body Tracking SDK, represents a further step beyond the previous Kinect versions [17]. The accuracy of skeletal joint extraction has been compared with a gold standard motion capture system and a high accuracy inertial measurement unit. Results confirm the high reliability of the vision device, which estimates the skeletal joints without significant discrepancies [18].

Although Azure Kinects are widely spread in research, they are mainly used singularly. For this reason, multi-camera systems based on Azure Kinects still need to be studied in depth. The development of multi-camera systems can often be the key to guaranteeing high reliability in several scenarios, from surveillance to manufacturing [19]. The peculiarity of multi-camera systems is mainly in the combination of features from different cameras, aiming to increase the reliability of the output data, which corresponds to a set of 3D coordinates. Of course, a configuration of multi-camera systems implies proper settings, mainly involving algorithms for calibration, which plays a critical role in many implementations. For instance, having two or more cameras calibrated is mandatory in several fields, mainly in manufacturing, where having a correct estimation of the position of humans, tools and/or robots is fundamental in reducing risks for the operators [20].

Calibration techniques allow interfacing between the 3D world and 2D camera images [21]. Referring to a single RGB-D camera based on time-of-flight technology, 3D points can be projected starting from 2D image coordinates, by knowing the intrinsic parameters of the camera and the estimated depth maps. If multiple cameras are considered, the calibration tries to minimize the inaccuracy of the 3D distance between homologous points, resulting from alignment errors in 2D images [22].

In the literature, several calibration methodologies for cameras with overlapping Field-of-Views (FOVs) cameras have been studied [23,24,25,26]. Most of them are based on the use of targets whose features can be robustly found. Such targets can be of one- [27], two- [28], or three-dimensions [29].

The calibration with 1D targets uses a minimum of three collinear points with known relative positioning. It must be noticed that camera calibration with such a pattern is pursued only if at least one point of the 1D object is fixed. The most considerable drawback of these calibration techniques resides in constructing the 1D pattern itself. In [30], a pattern with five black plastic spheres was used, with a diameter of 2 cm. The spheres were fixed on a metal stick. It is impossible to guarantee exact linearity among the points in this case. It could be challenging to extract the points of the calibration pattern (the centroids of the spheres), as can be done for more complex geometrical targets (2D or 3D objects).

There are several 2D patterns used for camera calibration, such as circle grids [31], Deltille grids [32], and Fiducial Marker Systems such as CALtag [33]. One of the most widely spread 2D targets is the chessboard [34], whose planar grid structure defines many points of a single image and makes such a target widely used in camera calibration, mainly because of its intuitive and straightforward configuration. The chessboard structure makes such a 2D target very easy to realize, as the only constraints to be respected are related to the size of the black and white squares and the rigidity of the chessboard itself, which must not show ripples and reliefs. In addition, several robust algorithms have been implemented to extract the key points from 2D chessboards, which make multi-camera calibration procedures easily applicable.

In the camera calibration using 3D patterns, a single 3D target, e.g., a sphere, is usually considered [35]. Although estimating the parameters for camera calibration using such a target is widely used in research, the precision of the method based on spheres is greatly influenced by the accuracy of ellipse fitting. Furthermore, it is not always easy to develop an object that satisfies all the characteristics meant for 3D patterns. The setup configuration for proper camera calibration can require time and effort, and the development of a suited three-dimensional object must meet specific conditions that are not immediate to follow. It should be noted that 3D objects can also be composed of multiple 1D patterns [36]. However, this kind of 3D calibration target leads to error mainly due to image noise and lighting conditions, which cause a sensible decrease in the accuracy in the feature extraction, particularly when compared to a planar pattern.

While the literature has mainly studied the camera calibration problem [37,38,39] for general cameras, the camera calibration task has poorly been addressed in the context of a network of Azure Kinect sensors. In [40], the authors estimate a coarse global registration among Azure Kinects, based on the data provided by the Body Tracking SDK algorithm. It is well known that such data bring an intrinsic error due to the estimation process pursued in the extraction of the skeletal joints [11]. This error inevitably affects the final registration, even if the authors try to refine the calibration methodology using a feature-matching algorithm. In general, it is clear that the knowledge of the physical characteristics of this low-cost RGB-D sensor allows for providing proper recommendations on how to specifically perform calibration in a context of multiple overlapping cameras, considering information extracted from either the inside RGB camera or the Infrared ones. Furthermore, addressing the calibration problem of RGB-D cameras without considering the raw depth information is strongly restrictive, as the presence of an IR sensor, along with the intrinsic functionalities of the Azure Kinect camera, can allow a highly accurate projection of the data in the 3D space.

This work compares different calibration methodologies and suggests a guideline of the best methods to properly calibrate multiple Azure Kinect cameras, according to the data that must be processed and the measures needed. The proposed methodologies all starts by analyzing a 2D target, i.e., a chessboard. This target is detected and processed in both RGB and IR images to estimate its corners. In a 3D approach, these points are projected in the 3D space, taking advantage of ToF principles. The chessboard becomes a “2.5D pattern” [41], as its planar features (corners) are directly computed from the depth map, using the intrinsic functionalities of the ToF RGB-D camera [42].

The main contributions of this work are:A two-camera system composed of Azure Kinects has been considered and the specific physical characteristics of these sensors have been studied to devise different calibration methodologies.Four different methodologies based on the data coming from color cameras and infrared cameras with or without the associated depth information have been compared in two real scenarios (dense point clouds of real objects for measures analysis, and people skeletal joints extracted from SDK Body tracking algorithm).A careful analysis of results provides a guideline for the best calibration techniques according to the element to be calibrated, i.e., point clouds with color or infrared resolutions and skeletal joints.

The paper is structured as follows. In Section 2, the proposed calibration methodologies are outlined. Section 3 defines the experimental setup in which the described calibration techniques are used. Section 4 analyzes the reliability of the proposed calibration methodologies applied to point clouds and skeletal joints. Finally, Section 5 and Section 6 draw a final discussion and the conclusion.

## 2. Methodology

The two-camera calibration methodologies discussed in this work consider Microsoft Azure Kinect. Such kind of device consists of an RGB camera and an infrared (IR) camera, with the latter providing depth information implementing ToF principles. Therefore, the Kinects output RGB images, IR images, and depth maps. The Azure Kinect is equipped with two software development kits for the management of all data that can be recorded by the internal cameras: the general Azure Kinect SDK and the Azure Kinect Body Tracking SDK [43]. In particular, the data provided by the Azure Kinect sensor can be represented in two different geometries: the geometry of the color camera or the geometry of the infrared camera. The term geometry, related to the RGB or IR sensors of the Azure Kinect camera, refers to a set of sensor properties, including the coordinate system, its resolution, and all intrinsic transformations. A set of routines in the general SDK allows the transformation of images or depth maps from one geometry to another. The Body Tracking SDK implements Deep Learning and Convolutional Neural Networks algorithms [44] to extract all the possible information for people segmentation, people tracking, and skeletal joint extraction. In Figure 1, the two cameras that produce RGB and IR images are shown. In the example images, the skeletal joints extracted by the body tracking SDK are superimposed.

Following the procedure in Figure 2, the depth maps acquired by the IR camera can be converted into point clouds by using the SDK functions [45]. Starting from the IR image, depth data can be converted directly, in the geometry of the infrared camera obtaining the $P^{\mathrm{infrared}}$ point cloud. Otherwise, the point cloud can be represented in the geometry of the color camera. In this case, the SDK provides a transformation $T_{\mathrm{intr}}$, that uses also the intrinsic camera parameters, to convert the depth map into a point cloud with color geometry. The result of this step is a $P^{\mathrm{color}}$ point cloud.

The proposed techniques consider a two-camera setup made of a Reference and a Template camera. Nevertheless, the system can be suited for multiple Azure Kinect calibration. Without any loss of generality, for multiple K cameras, the calibration has to be repeated (K−1) times to align the outputs of (K−1) Template cameras onto the Reference one. All calibration methodologies use a 2D target that will be captured simultaneously by the RGB and IR sensors of each of the two cameras. This target is a 2D chessboard made up of *m* rows and *n* columns of black and white squares with side length of *S*. The structured geometry of the chessboard guarantees robustness and accuracy for the corner detection and processing algorithms [46]. *F* frames of the chessboard are acquired by moving the target to different positions and orientations in the FOVs of both cameras.

The transformation matrix that relates the two coordinate systems of the Reference and Template Cameras is defined in the following Equation (Equation 1).
(1)$$T=\left(\begin{array}{cc} R & t\\ 0 & 1 \end{array}\right)$$
where $R\in R^{3\times 3}$ represents the rotation matrix and $t\in R^{3\times 1}$ the translation vector. The whole T matrix is estimated by evaluating the correspondences among the corners of the chessboard observed by the two cameras.

In Figure 3, the proposed calibration methodologies are graphically summarized. In this figure, $T_{\mathrm{intr}}$ and $T_{\mathrm{intr},\mathrm{Ref}}$ correspond to the intrinsic transformations that convert the data from the geometry of the infrared camera to the one of the color camera. On the other hand, four calibration matrices can be obtained comparing different camera sensors, namely RGB or IR sensors, and calibration procedures, namely 2D and 3D calibrations. Specifically, when chessboard corners are processed directly to estimate the transformation matrix, the calibration works with mere 2D image coordinates. Therefore, it ends with the following:$T^{2\mathrm{Dcolor}}$ if the chessboard corners are extracted from RGB images, i.e., with the geometry of the color cameras;$T^{2\mathrm{Dinfrared}}$ if the chessboard corners are detected in the IR images, i.e., with the geometry of the infrared cameras.

However, since the Azure Kinect computes depth maps of the environment, the same chessboard corners can be projected in 3D coordinates in the reference system of each camera. In this case, two further procedures working with 3D points can be defined to produce:$T^{3\mathrm{Dcolor}}$ if the chessboard corners are taken from RGB images and then projected in the 3D space, using the geometry of the color camera;$T^{3\mathrm{Dinfrared}}$ if the chessboard corners are extracted from the IR images and then projected in 3D, using the geometry of the infrared camera.

In the following subsections, the methodologies used to generate the 2D and 3D calibration matrices will be explained in detail.

### 2.1. 2D Calibration Procedures

A schematic pipeline of the 2D calibration methodology is shown in Figure 4.

Let ($I_{Ref}^{color}$, $I^{color}$) and ($I_{Ref}^{infrared}$, $I^{infrared}$) generically represent the images couples from the color and infrared sensors grabbed by the Reference and Template Azure Kinect cameras, respectively. The images are input to a corner detection algorithm [47] that estimates the 2D coordinates of the chessboard corners, namely (${\left(i_{c},j_{c}\right)}_{Ref}^{color}$,${\left(i_{c},j_{c}\right)}^{color}$) and (${\left(i_{c},j_{c}\right)}_{Ref}^{infrared}$, ${\left(i_{c},j_{c}\right)}^{infrared}$), with c={1,2,⋯,(m−1)(n−1)}. The corner coordinates from each of the *F* frames acquired during calibration, together with the square size *S* and the trial sets of intrinsic parameters for both cameras ($p_{0,\mathrm{Ref}}$ and $p_{0}$), feed the calibration algorithm, which finally estimates the intrinsic and extrinsic parameters of the camera [48]. The estimated intrinsic parameters include the focal length, the optical center, the skew, the Radial Distortion and the Tangential Distortion. This outcome refines the initial set of intrinsic parameters of both cameras. On the other hand, the extrinsic parameters define a rigid transformation to roto-translate the reference system of the Template camera into the reference system of the corresponding sensor of the Reference camera, as described in Equation (Equation 1). As depicted in Figure 4, the outputs of this 2D calibration procedure are $p_{\mathrm{Ref}}$, p, and the matrices $T^{2\mathrm{Dcolor}}$ or $T^{2\mathrm{Dinfrared}}$, depending on which sensor acquires the chessboard.

### 2.2. 3D Calibration Procedures

A schematic pipeline of the 3D calibration procedures is shown in Figure 5. Even in this case, the first step involves detecting the corner coordinates of the chessboard in the image reference system. The same methodology explained for 2D calibration returns again, for each frame acquired during calibration by the Reference and Template cameras, ${\left(i_{c},j_{c}\right)}_{Ref}^{color}$ and ${\left(i_{c},j_{c}\right)}^{color}$, or ${\left(i_{c},j_{c}\right)}_{Ref}^{infrared}$ and ${\left(i_{c},j_{c}\right)}^{infrared}$, depending on the considered sensor of the Azure Kinect. The 3D projection procedure converts the generic pixel coordinates (i,j) in world coordinates (x,y,z), defined in the corresponding reference system of the sensor. This transformation is performed at SDK level knowing the intrinsic parameters of both cameras $p_{0,\mathrm{Ref}}$ and $p_{0}$ (factory settings), and the corresponding depth maps. In particular, the latter is the result of the ToF measurement, performed by the IR sensor in its own geometry. The 3D projection generates points in 3D coordinates, namely ${\left(x_{c},y_{c},z_{c}\right)}_{Ref}^{color}$ and ${\left(x_{c},y_{c},z_{c}\right)}^{color}$, or ${\left(x_{c},y_{c},z_{c}\right)}_{Ref}^{infrared}$ and ${\left(x_{c},y_{c},z_{c}\right)}^{infrared}$. These points are the 3D positions of the chessboard corner, referred to in the geometries of the color and infrared cameras, respectively.

The 3D coordinates feed into the Maximum Likelihood Estimation Sample Consensus (MLESAC) estimator [49], which is a generalization of the Random Sample Consensus (RANSAC) algorithm [50]. RANSAC is an iterative method used for coordinate sets. In the first iteration, the algorithm selects random samples from the initial correspondences and finds the transformation matrix relative to the selected samples. This step is repeated iteratively, and the transformation returning the maximum number of matches, named inliers, is considered the optimal transformation matrix. All the other non-matched correspondences are considered outliers. One of the problems of the RANSAC algorithm is the setting of the threshold for correct matches. The MLESAC algorithm combines RANSAC with the Maximum Likelihood Estimation (MLE) method to find inliers. The goal of MLE is to find the optimal way to fit a distribution to the data [51]. By applying the MLE to the initial correspondences of each iteration, the noise dips are eliminated, thus excluding from the iterations those outliers that would be included if the samples were selected randomly. Hence, the estimate of the matching points provided by the MLESAC algorithm can be more precise and closer to the true solution, even requiring a reduced number of iterations to reach the optimal solution. In the specific case of interest, the MLESAC algorithm estimates the 3D transformation between the set of 3D points of the chessboard, collected from all the acquired frames. As a result, the calibration procedure determines the final calibration matrices $T^{3\mathrm{Dcolor}}$ and $T^{3\mathrm{Dinfrared}}$, as in Equation (Equation 1), depending on the sensor that acquires the chessboard images.

## 3. Experimental Setup

The real-case scenario in which experiments have been performed is shown in Figure 6. K=2 Azure Kinect sensors have been placed to have an extended overlapping area and the vision of the full body of people in the scene. The calibration methodologies have been evaluated in two different cases: (i) to assess the ability to reconstruct a target object by the combination of point clouds, and (ii) to estimate the robustness of the people’s skeleton alignment. Figure 7 shows the considered workspace grabbed by both Kinect sensors. The images show that the RGB camera has a field of view wider than that of the IR camera. In addition, the RGB camera has been set with a resolution of 3840 × 2160, while the IR camera has been set with a resolution of 640 × 576, to produce depth maps with narrow FOV [11].

In particular, two experiments have been performed:To state the capability of aligning point clouds, two analyses have been proposed: in a static scenario, a still object is placed in the two camera FOVs; in a dynamic scenario, a moving target is framed simultaneously by the two cameras. After the calibration phase, the point clouds in both infrared and color geometries, grabbed by the Template camera, are transferred into the coordinate system of the Reference.A subject stands still with open arms in front of the two cameras and the corresponding skeletal joints are extracted from the Azure Kinect Body Tracking library. The skeleton from the Template camera is transferred into the coordinate system of the Reference. In this case, ten consecutive frames have been collected to calculate the average position of each joint to reduce intrinsic errors [11] and average involuntary movements of the subject.

To have a clear visualization and avoid light reflections or color alterations of the chessboard due to the natural light or backlight effects, the workspace has been artificially illuminated using a light projector placed behind the Azure Kinects.

In the proposed configuration, the selected chessboard has m=6 rows and n=9 columns of black and white squares of side length S=45 mm. F=200 frames of the chessboard have been acquired. Figure 8 shows some examples of the RGB and IR images acquired during the experiments.

## 4. Calibration Analysis

The calibration methodologies have been evaluated considering the Root Mean Square Error (***RMSE***), defined as follows: (2)$$\mathrm{RMSE}=\sqrt{\frac{\sum_{j=1}^{J}{\left(d_{j}-\hat{d_{j}}\right)}^{2}}{J}}$$
where:$d_{j}$, $\hat{d_{j}}$ in the point cloud experiment are the 3D coordinates of points in correspondence taken from the Reference point cloud and the Template one after the application of estimated transformation. *J* is the total number of points in correspondence.$d_{j}$, $\hat{d_{j}}$ in the skeleton experiment are homologous 3D joint coordinates in the same reference system. Here, J=32 is the total number of the joints.

Low ***RMSE*** values indicate that points (or skeletal joints) are correctly transformed in the same reference system. The value of the ***RMSE*** has been calculated for each pair of point clouds and skeletons. Subsequently, the average of all RMSEs ($\bar{RMSE}$) and their standard deviation ($\sigma_{RMSE}$) were calculated to assess the proposed calibration techniques.

### 4.1. Point Cloud Experiment

Table 1 shows the quantitative results of the proposed calibration methodologies in the point cloud experiment considering a static target, i.e., a robot. Overall, 38 pairs of point clouds have been considered. RMSE values are computed comparing pairs of point clouds in the geometry of the color camera ($P^{\mathrm{color}}$ column) or in the geometry of the infrared camera ($P^{\mathrm{infrared}}$ column). Then, the mean of such values is computed, along with the standard deviation.

The mean of the RMSE values obtained in the alignment of the point clouds $P^{\mathrm{color}}$ demonstrate that the best calibration matrix is $T^{3\mathrm{Dcolor}}$, which produces an $\bar{RMSE}$ value equal to 21.426 mm. The worst result is obtained with the $T^{2\mathrm{Dinfrared}}$ matrix which provides an $\bar{RMSE}$ value of 37.283 mm. Even $\sigma_{RMSE}$ values confirm this analysis, since the variability of the RMSEs does not exceed 1 mm in any case. In addition, the calibration matrices that produce the lowest $\bar{RMSE}$s, also produce the lowest $\sigma_{RMSE}$.

Figure 9 provides a qualitative evaluation of the reconstructed point clouds in color geometry $P^{\mathrm{color}}$ obtained after the above calibrations. The images show the reconstruction of a static target, at 3.13 m from the Reference camera, resulting from the alignment of two point clouds considering the transformation matrices $T^{3\mathrm{Dcolor}}$ and $T^{2\mathrm{Dinfrared}}$. In the first case, the shape of the target is clearly visible, and its appearance is coherent and consistent with its expected shape. In the latter case, which underperforms the other calibrations, the target appears duplicated, and its 3D dense reconstruction fails.

In Table 1, the lowest value of $\bar{RMSE}$ calculated for the alignment of $P^{\mathrm{infrared}}$ point clouds in infrared geometry is 9.872 mm, obtained by $T^{3\mathrm{Dinfrared}}$, while the worst is 45.485 mm, obtained by $T^{2\mathrm{Dcolor}}$. Figure 10 shows the results of the alignment of the same static target of Figure 9, but modeled in the $P^{\mathrm{infrared}}$ point clouds in infrared geometry. Alignments are made by applying the best and worst calibration methodologies in Table 1. Specifically, the $T^{3\mathrm{Dinfrared}}$ calibration matrix produces a coherent reconstruction of the target, while the application of $T^{2\mathrm{Dcolor}}$ returns an altered version of the target shape, which seems shrunk in the front while its silhouette is not complete.

A careful analysis of the quantitative results of Table 1 highlights that the experiments carried out considering the calibration matrices resulting from the 2D calibration method give the worst results than the 3D calibration ones. The reason lies in the fact that $T^{2\mathrm{Dcolor}}$ and $T^{2\mathrm{Dinfrared}}$ are generated from matches between 2D data, while 3D calibration $T^{3\mathrm{Dcolor}}$ and $T^{3\mathrm{Dinfrared}}$ consider matches between sets of 3D coordinates that contain more information with the introduction of depth data. This result is not straightforward, since the computation of the depth maps, which is the basis of 3D calibration procedures, can suffer from implicit errors. However, such negative contributions do not influence 3D approaches, which always outperform 2D ones.

On the other side, it is possible to notice that the $P^{\mathrm{color}}$ presents the lowest $\bar{RMSE}$ values when using the $T^{3\mathrm{Dcolor}}$ calibration matrix, computed starting from the chessboard corners in RGB images. At the same time, the alignment of $P^{\mathrm{infrared}}$ point clouds in infrared geometry has the lowest $\bar{RMSE}$ with the calibration made by matching corners from IR images. These results can be explained considering the process that the Kinect sensor uses to produce the two point clouds in color or infrared geometries, as in Figure 2. The point clouds are always generated by the IR camera, but the transformation of the point cloud in the geometry of color camera requires an interpolation process that uses the intrinsic camera parameters. At the end of this process, the size of the point cloud greatly increases. In conclusion, the calibrations performed in the same space after the same transformations are those that perform better.

To better evaluate the proposed methodologies, the same calibration matrices have been applied to pairs of point clouds extracted from videos that frame a dynamic scene with a moving target. For this evaluation, 128 pairs of point clouds have been considered.

Table 2 shows the avarage and the standard deviation of the RMSEs obtained in comparing each couple of point clouds, in both color and infrared geometries. The results are highly comparable with the one observed in Table 1. The standard deviations show slightly higher values, as attributed to the presence of the moving target. Nevertheless, in all cases, $\sigma_{RMSE}$ values do not exceed 2.4 mm.

### 4.2. Skeleton Experiment

The $\bar{RMSE}$ values resulting from the comparison between the skeletal joints of the Template camera, aligned in the reference system of the Reference one for all the proposed procedures are reported in Table 3, together with the corresponding $\sigma_{RMSE}$ values. For such an evaluation, 15 pairs of skeletal joints have been aligned. Each pair contains the average values of the skeletal joints grabbed from both Template camera and Reference camera, performed within 10 frames. Hence, 150 frames have been considered in total. Observing the table, it is clear that the best result is obtained using the calibration matrix $T^{3\mathrm{Dinfrared}}$ with the lowest $\bar{RMSE}$ of 35.410 mm. The calibration performed using $T^{2\mathrm{Dcolor}}$, instead, gives the highest $\bar{RMSE}$ value, equal to 124.602 mm. As expected, the results are in accordance with those obtained for the point cloud in infrared geometry, shown in Table 1, since the skeletal joints are also generated in the IR environment, using the same IR camera of $P^{\mathrm{infrared}}$: the calibrations obtained in the same geometry produce a better overlap of the two skeletons.

The graphs reported in Figure 11 allow a qualitative evaluation of the effects of the best and worst calibration procedures on skeleton alignment. In Figure 11a, the results after the application of the $T^{3\mathrm{Dinfrared}}$ matrix are shown, while in Figure 11b the skeletons are registered using the $T^{2\mathrm{Dcolor}}$ matrix. The graphs confirm the results of the $\bar{RMSE}$ values. In Figure 11a the two skeletons are very close, while in Figure 11b some joints of the skeletons, especially those corresponding to the extreme joints of legs and arms, are very distant.

This result is very important if skeleton extraction is the target of a multi-camera setup. This goal is of increasing interest, since capturing humans from different points of view can lead to robust people tracking, even in case of camera occlusions and/or estimation errors. Furthermore, $\sigma_{RMSE}$ values confirm that the skeletal joints alignment is repeatable over the frames, as in all techniques they do not exceed 5.5 mm. Having a correct and continuous knowledge of where somebody is within a volume of interest is of critical importance to guarantee safety, for instance in human–robot collaboration, and even for action recognition tasks. In these scenarios, misalignment of the skeletons once referred to as a common coordinate system can lead to even huge and more dangerous errors. For this reason, performing a reliable camera calibration becomes mandatory.

## 5. Discussion

The camera calibration problem has been extensively addressed in the literature as the importance of having coherent data extracted from different sensors in a single reference system is widely recognized. However, with the availability of multi-modal sensors that provide different types of data, it is necessary to study calibration methods that take into account the specificity of the sensors and the type of data extracted. In this context, the presented work has filled the gap about the need for calibration methods specific to the Microsoft Azure Kinect cameras. Here, calibration methods have been developed starting from raw images from both the color and the infrared sensors. This choice guarantees a higher reliability in applying calibration to skeletons and point clouds, particularly with respect to [40]. Overall, the experiments have proven the efficiency of 3D-based techniques, which take advantage of the specifics of the Azure Kinect cameras. It must be noticed that the techniques here presented can be useful in calibrating a system composed of multiple Azure Kinects, as the alignment can be performed to indefinite pairs of point clouds and/or skeletons.

The main points of the proposed calibration methodologies are the following:In general, 3D procedures outperform 2D ones as depth information is added to the calibration. This is due to the effectiveness of depth estimation and intrinsic transformations used to project 2D image points in the 3D space.The alignment of point clouds in the geometry of the color camera has the lowest error value when using a calibration procedure working in 3D starting from RGB images, since both the point cloud in color geometry and the chessboard corner coordinates enabling the calibration follow the same interpolation procedures carried out by the general SDK functions.The alignment of point cloud in the geometry of the infrared camera has the lowest error when the calibration works starting from IR images. Even in this case, the calibration performed in the same geometry of the point cloud produces the best result.The alignment of skeletons shows the best result while calibration is performed in 3D starting from IR images. It further confirms the previous statement.In all experiments, the standard deviations of the RMSE values state that the variability in error computations is always lower than the improvement in aligning both point clouds and skeletal joints.

The results are significant in systems with two or more cameras, mainly when low-cost sensors, such as Azure Kinects, can be efficiently used for several applications to have full 3D representations of targets and environments. For example, 3D characterization is helpful in many computer vision applications, such as 3D reconstruction, 3D localization, and 3D pose estimation. Building a proper 3D scene can allow a highly accurate assessment of a 3D map for pursuing, for instance, the reconstruction of an industrial object. Furthermore, estimating human 3D movements is required in various scenarios, which may need to detect specific activities performed by the framed subjects. To segment and recognize human movements, a properly calibrated camera system can provide a complete reconstruction of human posture, overcoming any occlusion that may limit the view of one of the cameras. Such calibration processes can be useful in surveillance, where it is crucial to know what a person is doing and where he/she is going. Furthermore, a calibrated system that provides a complete set of 3D skeletal joints, or a dense point cloud, can easily represent a subject executing a specific task. Such data may widely facilitate segmentation and, thus, recognition of the actions needed for any assignment.

## 6. Conclusions

In this paper, a two-camera system composed of Azure Kinect sensors has been considered. Starting from the possible data that can be extracted by these sensors (color and infrared images, point clouds in the geometry of the color camera, and point clouds in the geometry of the infrared camera), four different calibration procedures working in the 2D or 3D spaces have been compared. The analysis of results in two real-case scenarios has provided a guideline to properly calibrate dense point clouds or skeletal joints according to the geometry in which they are represented.

A future step in the calibration may involve multi-camera calibration. Even though the proposed systems can be applied to a multiple Azure Kinect system, the methodologies always consider an in-pair calibration of the cameras, i.e., Reference-Template camera calibration. It would be interesting to study the calibration of multiple Azure Kinects at the same time, using global optimization algorithms. The further redundancy of a multi-camera will be the key to better estimations of dense point clouds and occlusion-robust skeletons.

## Figures and Tables

**Figure 1 sensors-22-04986-f001:**
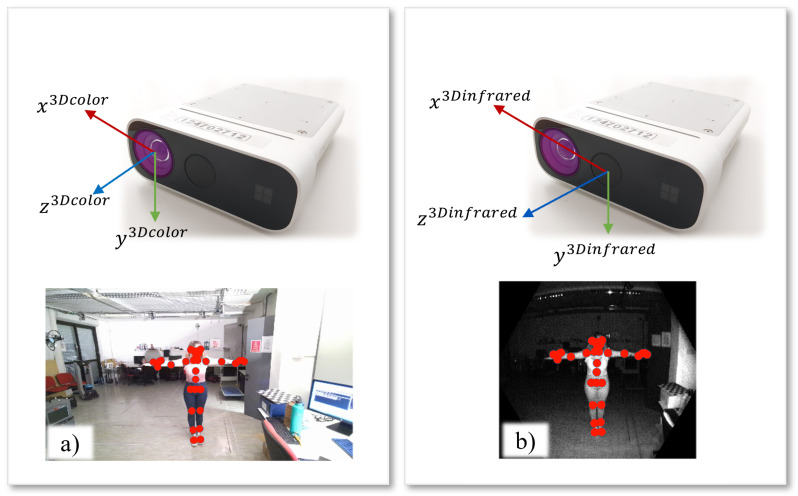
Representation of the internal Kinect sensors that produce: (**a**) color images with a resolution of 3840 × 2160 and (**b**) IR images with resolution 640 × 576. The origin of the coordinate systems is placed at the focal point of each sensor [11]. The skeletal joints extracted by the Body Tracking SDK are superimposed on the images.

**Figure 2 sensors-22-04986-f002:**
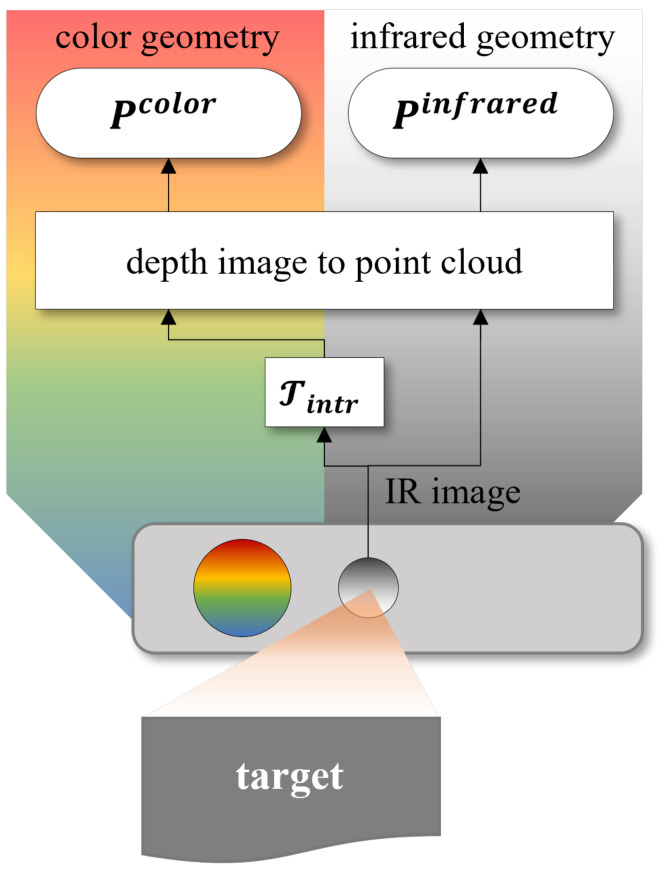
Schematic representation of the point cloud realization with color and infrared geometries, using the Azure Kinect SDK.

**Figure 3 sensors-22-04986-f003:**
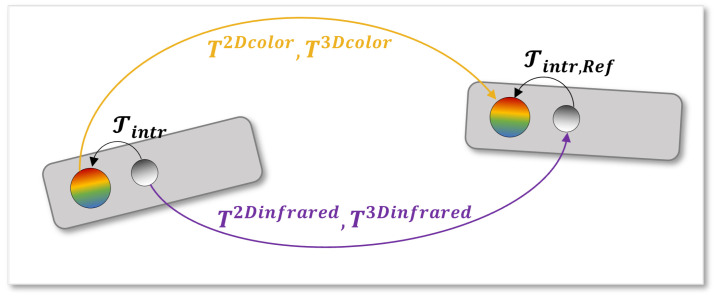
Conceptual meaning of the application of the transformation matrices obtained from the proposed calibration techniques. The transformation process using matrices with color geometry is marked in yellow, whereas the transformation process using matrices with infrared geometry is marked in purple.

**Figure 4 sensors-22-04986-f004:**
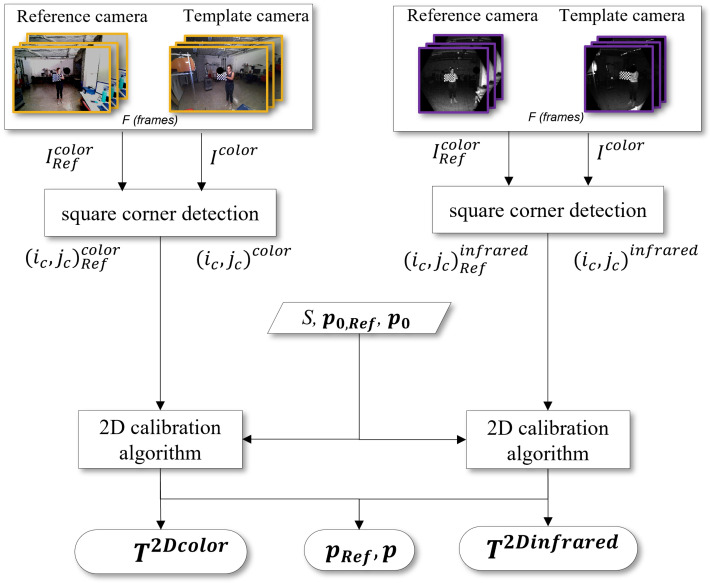
2D calibration flow chart for the creation of the transformation matrices $T^{2\mathrm{Dcolor}}$ and $T^{2\mathrm{Dinfreared}}$. The frames with color resolution are marked in yellow, while the frames with infrared resolution are marked in purple.

**Figure 5 sensors-22-04986-f005:**
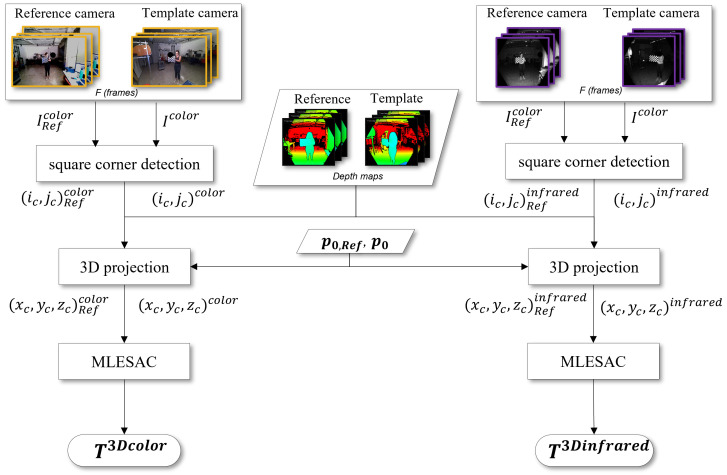
3D calibration flow chart for the creation of the transformation matrices $T^{3\mathrm{Dcolor}}$ and $T^{3\mathrm{Dinfrared}}$. The frames with color resolution are marked in yellow, while the frames with infrared resolution are marked in purple.

**Figure 6 sensors-22-04986-f006:**
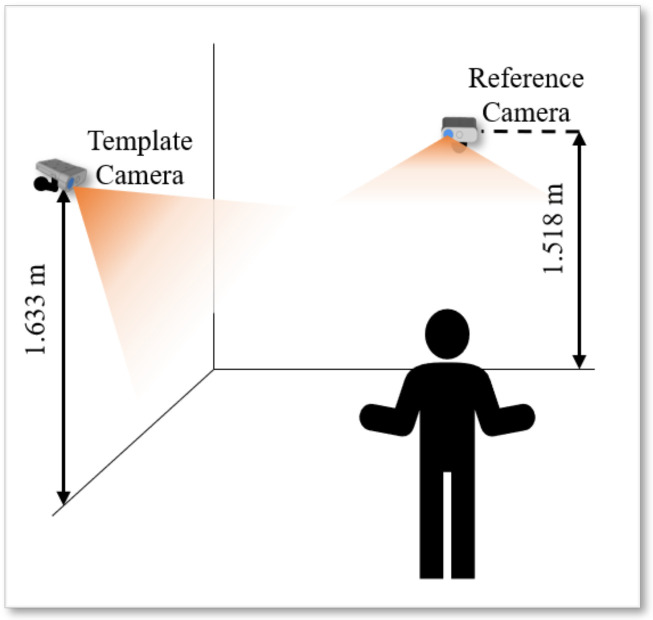
Depiction of the experimental setup considering two Azure Kinects.

**Figure 7 sensors-22-04986-f007:**
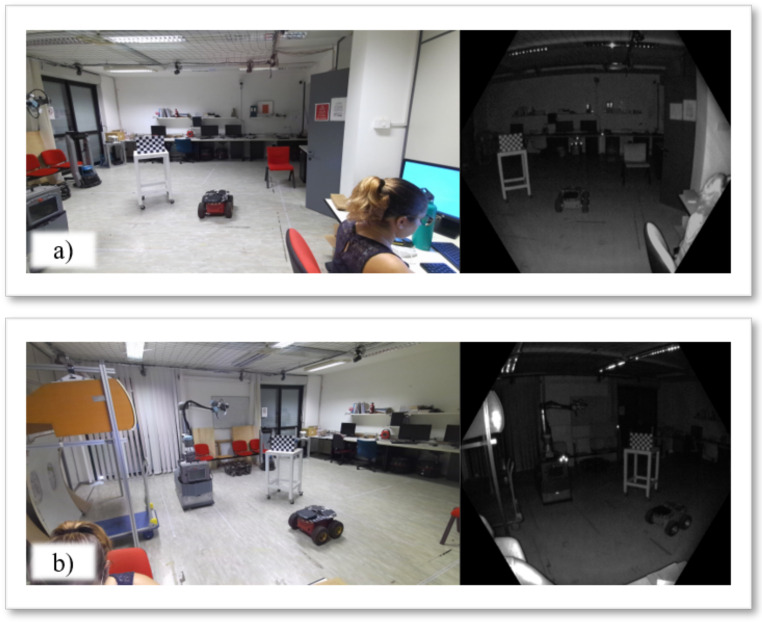
Views of the two Azure Kinects used during the experimentation, where (**a**) represents the Reference Camera, and (**b**) represents the Template Camera. More specifically, the images on the left in both (**a**,**b**) show the frames grabbed from the RGB sensors, while the images on the right show the frames grabbed from the IR sensors.

**Figure 8 sensors-22-04986-f008:**
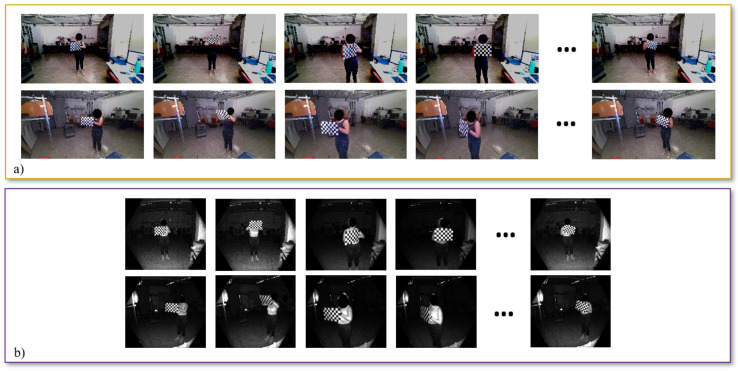
(**a**) RGB and (**b**) IR image samples of the chessboard. Several positions and orientations have been considered to optimize the results of the calibration.

**Figure 9 sensors-22-04986-f009:**
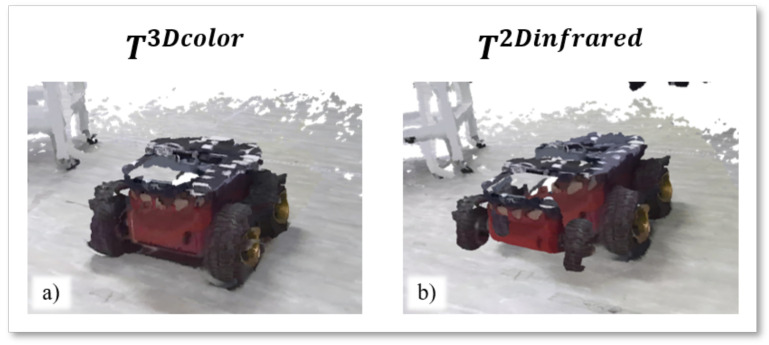
Visual representation of the (**a**) best ($T^{3\mathrm{Dcolor}}$) and (**b**) worst ($T^{2\mathrm{Dinfrared}}$) alignment of point cloud in color geometry $P^{\mathrm{color}}$. The input point clouds are captured at the same timestamp from both the Azure Kinect cameras.

**Figure 10 sensors-22-04986-f010:**
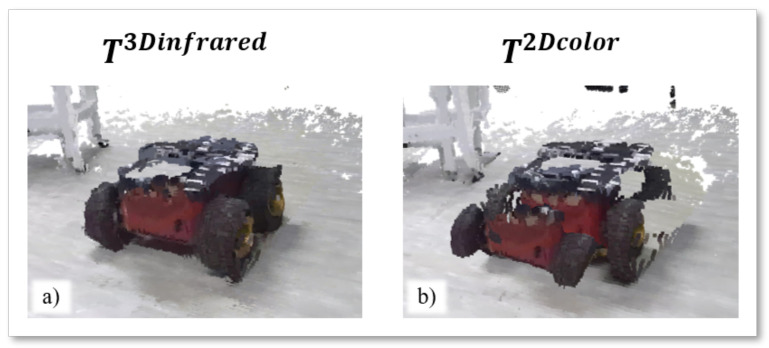
Visual representation of the (**a**) best ($T^{3\mathrm{Dinfrared}}$) and (**b**) worst ($T^{2\mathrm{Dcolor}}$) alignment of $P^{\mathrm{infrared}}$ point clouds. he input point clouds are captured at the same timestamp from both the Azure Kinect cameras.

**Figure 11 sensors-22-04986-f011:**
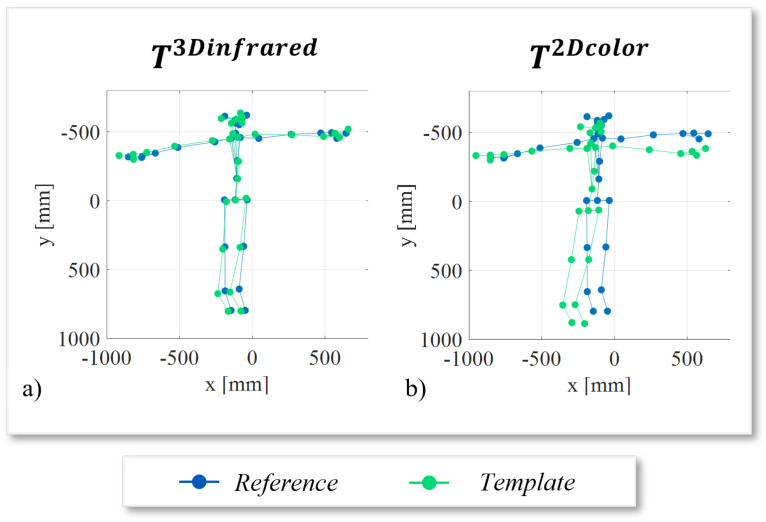
Graphic representation of the (**a**) best ($T^{3\mathrm{Dinfrared}}$) and (**b**) ($T^{2\mathrm{Dcolor}}$) worst skeleton alignments. In both graphs, the aligned template skeleton is in green, while the reference skeleton is in blue.

**Table 1 sensors-22-04986-t001:** Mean and standard deviation of RMSEs calculated between aligned and reference point clouds with static target, where calibration techniques have been applied [mm]. The best results of $\bar{RMSE}$ are underlined.

	$P^{\mathrm{color}}$		$P^{\mathrm{infrared}}$	
	$\bar{\mathrm{RMSE}}$	$\sigma_{\mathrm{RMSE}}$	$\bar{\mathrm{RMSE}}$	$\sigma_{\mathrm{RMSE}}$
$T^{2\mathrm{Dcolor}}$	24.427	0.918	45.485	0.804
$T^{2\mathrm{Dinfrared}}$	37.283	0.955	20.162	0.592
$T^{3\mathrm{Dcolor}}$	21.426	0.608	36.833	0.735
$T^{3\mathrm{Dinfrared}}$	33.194	0.758	9.872	0.268

**Table 2 sensors-22-04986-t002:** Average and standard deviation of the RMSEs calculated between aligned and reference point clouds with dynamic target, where the calibration techniques have been applied [mm]. The best results of $\bar{RMSE}$ are underlined.

	$P^{\mathrm{color}}$		$P^{\mathrm{infrared}}$	
	$\bar{\mathrm{RMSE}}$	$\sigma_{\mathrm{RMSE}}$	$\bar{\mathrm{RMSE}}$	$\sigma_{\mathrm{RMSE}}$
$T^{2\mathrm{Dcolor}}$	25.340	0.666	36.683	2.383
$T^{2\mathrm{Dinfrared}}$	39.446	2.299	13.046	0.765
$T^{3\mathrm{Dcolor}}$	20.868	1.233	33.122	2.198
$T^{3\mathrm{Dinfrared}}$	34.039	2.024	7.429	0.606

**Table 3 sensors-22-04986-t003:** Average and standard deviation of the RMSEs calculated between aligned and reference skeletal joints, where calibration techniques have been applied [mm]. The best result of $\bar{RMSE}$ is underlined.

	Joints of the Skeleton	
	$\bar{\mathrm{RMSE}}$	$\sigma_{\mathrm{RMSE}}$
$T^{2\mathrm{Dcolor}}$	124.602	1.349
$T^{2\mathrm{Dinfrared}}$	44.256	4.640
$T^{3\mathrm{Dcolor}}$	111.247	1.889
$T^{3\mathrm{Dinfrared}}$	35.410	5.490

## Data Availability

Not applicable.

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
