# Peer review of "Microsoft Azure Kinect Calibration for Three-Dimensional Dense Point Clouds and Reliable Skeletons"

_sensors, 2022, doi:10.3390/s22134986_

Round 1
Reviewer 1 Report
The article is very good. I just recommend the authors to improve the conclusions based on the achievements presented in the manuscript and to conduct a proofreading on the text to remove any errors and flaws.
Author Response
We would like to thank the Reviewer for his/her feedback about the presented work. To better enlighten the conclusions, we split the discussion and the conclusions into two different sections.
Besides, the discussion section reports a new outcome of a variability study, showing that the improvements in the point cloud and skeleton alignment over multiple data pairs are always more significant than the standard deviation of the computed RMSEs.
Furthermore, as the Reviewer suggested, the manuscript has been proofread carefully, aiming to check for typos and errors.
Reviewer 2 Report
I reviewed the paper in detail and the paper needs a few revisions to be published.
1. A quantitative evaluation of the results should be given at the end of Abstract.
2. Discussions should be a separate section and should be extended to reflect a solid comparison (advantages / disadvantages /limitations) of the proposed method over state of the art methods.
3. Conclusion, should be re-designed to a single paragraph (can be already extracted from current Discussions and conclusions)
Author Response
We want to thank the Reviewer for the relevant comments about our work.
As suggested, we add a quantitative evaluation of the results in the Abstract. Specifically, the last part of the Abstract now reads:
Experiments demonstrate that the best results are returned by applying 3D calibration procedures, which give an average distance between all couples of corresponding points of point clouds in color or infrared resolution of 21.426 mm and 9.872 mm for a static experiment and of 20.868 mm and 7.429 mm while framing a dynamic scene. At the same time, the best results in body joint alignment are achieved by three-dimensional procedures on images captured by the infrared sensors, resulting in an average error of 35.410 mm.
The Discussion and the Conclusions have been split into two sections.
- The Discussion has been extended by adding a few comments on the achieved higher reliability than the one of Lee et al. (Sensors, 2021) due to the use of 3D-based techniques, specifically developed for the Azure Kinect sensor. Moreover, the discussion summarizes the result of a newly-added variability study. This analysis showed that the improvements in the point cloud and skeleton alignment over multiple data pairs were always more significant than the standard deviation of the computed RMSEs.
- Conclusions have been re-designed in two short paragraphs: the first on the achieved results and the second on future work. We do believe that this arrangement can help the reader understand the final message of our work.
Finally, the manuscript has been proofread carefully, aiming to check for typos and errors.
Reviewer 3 Report
The paper compares different methods for multiple Microsoft Azure Kinect calibration. Although the paper is well written and the topic is of a great interest, results should be improved. In particular, for each calibration approach, the evaluation is made considering the RMSE. However, i've understand that this comes out from a single experiment for each technique. I suggest to carry out more tests in order to better suppoort the final verdict. For instance, i suppose that for each technique by carrying out different tests one would obtain different values of RMSE. If these values oscillate in a small range maybe the verdict will be the same. But what if they fall in a wider range with high uncertainty?
Author Response
We sincerely thank the Reviewer for the interesting comments and suggestions.
As suggested, experiments have been re-designed to prove better the validity of calibration approaches. Specifically, we considered more pairs of frames from the two cameras instead of single pairs.
As a first step, the point cloud analysis is now carried out by framing a static or a dynamic scene.
In the static case, 38 pairs of point clouds from an extended video were collected and aligned using the same four calibration matrices obtained by the proposed methodologies. Then, the average and the standard deviation of the RMSE were evaluated for the final assessment. Specifically, the mean values of RMSEs were highly comparable with the results shown in the previous version of the manuscript, whereas the standard deviations of RMSEs did not exceed 1 mm.
In the dynamic case, 128 pairs of point clouds modeling a scene with a moving target were considered and aligned following the same approaches. The tests have shown results comparable to those obtained in the static experiments, with a maximum variability (standard deviation of RMSEs) of about 2.4 mm.
In addition, more experiments have been carried out on a more significant number of skeletons. More specifically, 15 sets of skeletal joints were considered, where each joint is placed at the average position computed on 10 consecutive frames. The mean values of the RMSEs after skeleton alignment agreed with the RMSEs of the previous manuscript version, whereas the standard deviations of RMSEs did not exceed 5.5 mm
In any case, the standard deviations are below the improvements of RMSE, thus supporting the final verdict and the repeatability of the tests.
The manuscript has been modified accordingly in lines 277-292 and 328-335, while Tables 1-3 have been completely revised. Even the discussion shows a new point which reads:
In all experiments, the standard deviations of the RMSE values state that the variability in error computations is always lower than the improvement in aligning both point clouds and skeletal joints.
Finally, the manuscript has been proofread carefully, aiming to check for typos and errors.